# A Statistical Mechanics Framework for
# Task-Agnostic Sample Design in Machine Learning

**Bhavya Kailkhura**[1], **Jayaraman J. Thiagarajan**[1], **Qunwei Li** [2], **Jize Zhang**[1],
**Yi Zhou**[3], **Peer-Timo Bremer**[1]
[1]Lawrence Livermore National Laboratory, [2]Ant Financial, [3]The University of Utah
{kailkhura1,jjayaram,zhang64,bremer5}@llnl.gov,qunwei.qw@antfin.com,yi.zhou@utah.edu

## Abstract

In this paper, we present a statistical mechanics framework to understand the effect of sampling properties of training data on the generalization gap of machine learning (ML) algorithms. We connect the generalization gap to the spatial properties of a sample design characterized by the pair correlation function (PCF). In particular, we express generalization gap in terms of the power spectra of the sample design and that of the function to be learned. Using this framework, we show that space-filling sample designs, such as blue noise and Poisson disk sampling, which optimize spectral properties, outperform random designs in terms of the generalization gap and characterize this gain in a closed-form. Our analysis also sheds light on design principles for constructing optimal task-agnostic sample designs that minimize the generalization gap. We corroborate our findings using regression experiments with neural networks on: a) synthetic functions, and b) a complex scientific simulator for inertial confinement fusion (ICF).

## 1  Introduction

Machine learning (ML) techniques have led to incredible advances in a wide variety of commercial applications, and similar approaches are rapidly being adopted in several scientific and engineering problems. Traditionally, ML research has focused on developing modeling techniques and training algorithms to learn generalizable models from historic labeled data (i.e., a known set of inputs and their corresponding responses). However, in several applications, we encounter a key challenge even before building the model – determining the input samples for which the responses should be collected (referred to as *task-agnostic sample design* problem). This is particularly true for emerging applications in physical sciences and engineering where curated datasets are not available *a priori* and data acquisition involves time-consuming computational simulations or expensive real-world experiments. For example, in inertial confinement fusion (ICF) [2], one needs to build a high-fidelity mapping from the process inputs, say target and laser settings, to process outputs, such as ICF implosion neutron yield and X-ray diagnostics. In such scenarios, the properties of collected data directly control the generalization error of ML models. However, determining the right samples to use for model training hinges on understanding the intricate interplay between sampling properties and the ML generalization error. Unfortunately, our theoretical understanding is very limited in this regard, and hence existing sample design approaches rely upon a variety of heuristics, e.g., generating so called space-filling sample designs [15] to cover the input space as uniformly as possible.

Most existing theoretical frameworks only study the generalization properties of random i.i.d. designs or other simple probabilistic variants. Intuitively, this assumption ignores the dependency of generalization gap on data properties except the sample size (data-independent bounds). While some efforts exist to obtain data-dependent bounds, they still focus on studying model design related questions while ignoring sample design aspects. To the best of our knowledge, there does not exist

a framework in the literature that can help study the generalization error of generic sample designs (e.g., space-filling). This paper proposes to study generalization error from the viewpoint of the sampler generating the training data. We fill a crucial gap by developing a framework capable of characterizing the generalization performance of generic sample designs based on metrics from statistical mechanics, which are expressive enough to quantify a broad range of sample distributions.

**Contributions**: We develop a framework for studying the generalization behavior of sample designs through the lens of statistical mechanics. First, we model sample design as a stochastic point process and obtain a corresponding representation in the spectral domain using tools from [18]. This approach allows us to study the behavior of a larger class of sample designs (including space-filling). In particular, for our subsequent analysis, we focus on the blue noise [13, 17] and Poisson disk sampling (PDS) [16, 18] designs (see Figure 1 in the supplementary material). Next, we reformulate the generalization gap in the spectral domain and obtain an explicit closed-form relation of generalization gap with the power spectra of both the sample design and the function to be learned. Using our framework, we are able to theoretically show that space-filling designs outperform random designs. We further characterize the gains obtained with two state-of-the-art space-filling designs, namely blue noise and PDS samples, over a random design in a closed-form. This analysis further enables us to formulate design principles to construct optimal sampling methods for specific ML problems. We also make interesting (counter-intuitive) observations on the convergence behavior of generalization error with increasing dimensions. Specifically, we find that analysis with traditional metrics leads to inconsistent results in high dimensions. To overcome this issue, we develop novel spectral metrics to obtain meaningful convergence results for different sampling patterns (see supplementary material). Finally, we corroborate our findings by carrying out regression experiments on synthetic functions and a complex scientific simulator for inertial confinement fusion [3].

## 2   Related Work

### 2.1   Generalization Theory

Understanding the generalization error is essential for estimating how well the generated hypothesis will apply to unknown test data. Traditionally, generalization error is analyzed based on the model complexity, such as, the Vapnik-Chervonenkis (VC) dimension and the Rademacher complexity [4], or properties of the learning algorithm, such as uniform stability [5], and data-independent upper bounds on the error are derived. Some efforts have extended these bounds to accommodate certain data-related properties – for e.g., the luckiness framework [19], empirical Rademacher complexity [14], and the robustness of learning algorithms [27]. However, most existing frameworks are either restricted to random i.i.d. designs or cannot accommodate a broad range of sample designs. Consequently, they cannot be leveraged to gain insights into obtaining improved sample designs.

### 2.2   Space-Filling Designs

Sample design has been a long-standing research area in statistics [11, 22]. Traditionally, a good task-agnostic sample design aims to uniformly cover the input space to generate the so-called *space-filling* designs [15]. Since it is challenging to evaluate the space-filling property, simple scalar metrics, e.g., discrepancy [7] or geometric distances (maximin or minimax [24]) are utilized. However, these scalar metrics are not very descriptive, and when used as the design objective, often result in poor-quality samples. Recent work in [18] overcame this limitation using a spectral framework to quantify the space-filling property and demonstrated superiority over other designs. However, these strategies are not designed to specifically improve the generalization error of learning algorithms and, more importantly, it is currently not possible to rigorously characterize their generalization performance.

### 2.3   Other Related Directions

The term sample design is used broadly, and can refer to a variety of problems including subset selection [8, 1], linear bandits [9], diversity sampling [20] and active learning [25]. The fundamental difference between these works and the setup considered in this paper is that our sample design process is agnostic to both the specific response (i.e., output) and the choice of the ML model, thus, referred to as task-agnostic sample design.

# 3 Preliminaries – A Statistical Mechanics View of Sample Design

Studying the effect of sample design on the generalization error requires the use of expressive metrics to characterize sampling properties. Sample designs have been traditionally analyzed using heuristic measures, such as, discrepancy or uniformity which are known to be insufficient [18]. Hence, we advocate the use of a principled statistical mechanical analysis where a sample design is modeled as a stochastic point process and characterized using both its power spectral density (PSD) in spectral domain and pair correlation function (PCF) in spatial domain.

**Power Spectral Density**: For a sample design with finite set of $N$ samples, $\{\mathbf{x}_j\}_{j=1}^N$, the PSD describes how signal power is distributed over frequencies $\mathbf{k}$. It is formally defined as

$$P(\mathbf{k}) = \sum_{j,\ell} e^{-2\pi i \mathbf{k} \cdot (\mathbf{x}_\ell - \mathbf{x}_j)} / N.$$

For isotropic designs, $P(\rho) = P(|\mathbf{k}|)$ where $\rho$ is the radial frequency and $|\cdot|$ is the magnitude operator. Equivalently, a sample design can also be characterized in the spatial domain.

**Pair Correlation Function**: For a sample design, the PCF describes how sample density varies as a function of distances $\mathbf{r}$. For isotropic designs, $G(r) = G(|\mathbf{r}|)$ where $r$ is the radial distance.

**Relating PCF and PSD**: The PSD and PCF of a sample design are related via the Fourier transform as follows:

$$
\begin{aligned}
P(\mathbf{k}) &= 1 + NF\left(G(\mathbf{r}) - 1\right) \\
&= 1 + N \int_{\mathbb{R}^d} (G(\mathbf{r}) - 1) \exp(-2\pi i \mathbf{k}.\mathbf{r}) d\mathbf{r},
\end{aligned}
\tag{1}
$$

where $F(\cdot)$ denotes the $d$-dimensional Fourier transform. For isotropic designs (focus of this paper), the above relationship simplifies as follows.

**Theorem 1.** *The PCF and the PSD of radially symmetric sample designs are related as follows:*

$$G(r) = 1 + \frac{1}{N} r^{1-\frac{d}{2}} H_{\frac{d}{2}-1}\left(\rho^{\frac{d}{2}-1}(P(\rho) - 1)\right),$$

*where $H_d(f(\rho)) = 2\pi \int_0^\infty \rho f(\rho) J_d(2\pi r \rho) d\rho$ denotes the Hankel transform and $J_d(.)$ is the Bessel function of order $d$.*

**Realizability**: Note, not all PSDs/PCFs are realizable in practice. The two necessary conditions [1] that a sample design must satisfy to be realizable are: (a) its PSD must be non-negative, i.e., $P(\mathbf{k}) \geq 0, \ \forall \mathbf{k}$, and (b) its PCF must be non-negative, i.e., $G(\mathbf{r}) \geq 0, \ \forall \mathbf{r}$.

Theorem 1 along with realizability conditions establish a fundamental relationship between the PSD and PCF of isotropic sample designs. We utilize this to construct optimal forms of PDS and blue noise (Lemmas 1, 2, 8 and 9) and carry out our analysis only on realizable power spectra. Note that not every power spectra has corresponding sample design, though other way around is always true.

# 4 Risk Minimization using Monte Carlo Estimates

We consider the following general supervised learning setup: We have two spaces of objects $X \in \mathbb{T}^d$ (toroidal unit cube $[0,1]^d$) and $Y \in \mathbb{R}$ where $Y = \mathcal{F}(X)$. The goal of a learning algorithm is to learn a function $h : X \to Y$ (often called *hypothesis*) which approximates the true (but unknown) function $\mathcal{F}$. We assume access to training data comprised of $N$ samples $S = \{(x_1, y_1), \cdots, (x_N, y_N)\}$ drawn from an unknown distribution $P(x, y)$. We infer a hypothesis $h(\cdot)$ by minimizing the population risk:

$$R_P(h) \triangleq \mathbb{E}_{P(x,y)}[l(h(x), y)] = \int l(h(x), y) dP(x, y),
\tag{2}$$

where $l(\cdot, \cdot)$ denotes the loss function.

**Empirical Risk Minimization**: In general, the joint distribution $P(x, y)$ is unknown to the learning algorithm and hence the risk $R_P(h)$ cannot be computed. Instead, an approximation referred as *empirical risk* is often used which is obtained by averaging the loss function on the training data:

$$R_S(h) \triangleq \frac{1}{N} \sum_{i=1}^{N} l(h(x_i), y_i). \tag{3}$$

Note that the empirical risk $R_S(h)$ is a Monte Carlo (MC) estimate of the population risk $R_P(h)$. It also can be rewritten in a continuous form

$$R_S(h) \triangleq \frac{1}{N} \int_{\mathbb{T}} S(x) l(h(x), y) dx, \tag{4}$$

where $\mathbb{T}$ is the sampling domain and $S(x)$ is the sampling function, i.e., a sample design rewritten as a random signal $S$ composed of $N$ Dirac functions at positions $S(x) = \sum \delta(x - x_i)$ for $i = 1, \cdots, N$.

**Generalization Gap**: In ML and statistical learning theory, the performance of a supervised learning algorithm is measured by the generalization gap, which is the expected difference between the population risk and the empirical risk. More specifically, we adopt the following definition of the generalization gap:

$$\text{gen}(h) \triangleq \mathbb{E}_S[(R_P(h) - R_S(h))^2], \tag{5}$$

which is the expected difference between the population risk and its empirical risk on the training data for a fixed hypothesis $h(\cdot)$[2]. The generalization gap also has an alternate form with a direct link to the statistical properties of the sampling pattern:

$$\begin{aligned}
\text{gen}(h) &\triangleq \mathbb{E}_S[(R_P(h) - R_S(h))^2] \\
&= bias^2 + var(R_S(h)).
\end{aligned}$$

We consider sample designs which are homogeneous, i.e. statistical properties of a sample are invariant to translation over the sampling domain. Homogeneous sample designs are unbiased in nature, thus, the generalization gap arises only from the variance. Though variance analysis of Monte-Carlo integration has been considered in the literature [10, 26, 23], such an analysis has not been carried out so far, in the context of the generalization gap in ML.

## 5 Connecting Generalization Gap with Sample Design

### 5.1 Monte Carlo Estimator of Risk in the Spectral domain

Building upon [23], the MC estimator for risk as given in Eq. 4 can be transformed to the Fourier domain $\phi$ using the fact that the dot-product of functions (the integral of the product) is equivalent to the dot-product of their Fourier coefficients. This allows us to pose the MC estimator for empirical risk as follows:

$$R_S(h) \triangleq \frac{1}{N} \int_{\phi} \mathcal{F}_S(\mathbf{k}) \mathcal{F}_l(\mathbf{k})^* d\mathbf{k}, \tag{6}$$

where $\mathcal{F}_S$, $\mathcal{F}_l$ denote the Fourier transforms of the sampling function $S$ and the loss function $l$ and $(*)$ denotes complex conjugate.

### 5.2 Spectral Analysis of the Generalization Gap

We now use the spectral domain version of empirical risk to define the generalization gap:

$$\begin{aligned}
\text{gen}(h) &\triangleq bias^2 + var(R_S(h)) = (\mathbb{E}(R_S(h)) - R_P(h))^2 + \mathbb{E}(R_S(h)^2) - (\mathbb{E}(R_S(h)))^2 \\
&= (\mathbb{E}(R_S(h)) - R_P(h))^2 + \frac{1}{N^2} \int_{\phi \times \phi} \mathbb{E}(\mathcal{F}_{S,l}(\mathbf{k}, \mathbf{k}')) d\mathbf{k} d\mathbf{k}' - (\mathbb{E}(R_S(h)))^2, \tag{7}
\end{aligned}$$

where $\mathcal{F}_{S,l}(\mathbf{k}, \mathbf{k}') \triangleq \mathcal{F}_S(\mathbf{k}) \cdot \mathcal{F}_l(\mathbf{k})^* \cdot \mathcal{F}_S(\mathbf{k}')^* \cdot \mathcal{F}_l(\mathbf{k}')$. Using this definition, we derive an explicit closed-form relation of the generalization gap with the power spectra of both $S$ and $l$. To this end, we first simplify Eq. 7 by restricting our analysis to homogeneous designs:

**Theorem 2.** *The generalization gap for homogeneous sample designs in terms of the power spectra of both the sampling pattern $\mathcal{P}_S$ and the loss function $\mathcal{P}_l$ can be obtained as:*

$$gen(h) \triangleq \frac{1}{N} \int_\Theta \mathbb{E}(\mathcal{P}_S(\mathbf{k}))\mathcal{P}_l(\mathbf{k})d\mathbf{k}, \tag{8}$$

*where $\Theta$ is the Fourier domain $\phi$ without DC frequency.*

By combining Theorem 2 with Eq. 5, one can calculate the generalization gap of arbitrary sample designs in terms of their power spectra. When the design is isotropic (i.e., the power spectrum is radially symmetric), the error can be directly computed from the radial mean power spectra of the loss with $\rho = |\mathbf{k}|$, i.e., $\hat{\mathcal{P}}_l$ and the sample design $\hat{\mathcal{P}}_S$.

**Proposition 3.** *The generalization gap for isotropic homogeneous sample designs is*

$$gen(h) \triangleq \frac{\mu(\mathcal{S}^{d-1})}{N} \int_0^\infty \rho^{d-1}\mathbb{E}(\hat{\mathcal{P}}_S(\rho))\hat{\mathcal{P}}_l(\rho)d\rho, \tag{9}$$

*where $\mu(\mathcal{S}^d)$ is the Lebesgue measure of a $d$-dimensional unit sphere in $\mathbb{R}^d$ given by $2\sqrt{\pi^d}/\Gamma(d/2)$.*

*Proof.* These results can be obtained by rewriting Theorem 2 in polar coordinates and noting that the power spectra is radially symmetric for isotropic functions. $\square$

## 6 Best and Worst Case Generalization Gap

The proposed framework requires us to explicitly know the power spectra of the loss function to calculate the generalization gap, which is usually unknown. Hence, in this section, we restrict our analysis to a particular class of integrable functions of the form $l(x)_{\mathcal{X}_\Omega}$ with $l(x)$ smooth and $\Omega$ a bounded domain with a smooth boundary where $\mathcal{X}_\Omega$ is the characteristic function of $\Omega$ (see [6] for more details). We consider a best-case function and a worst-case function, both from this class to quantify the generalization behavior over the entire class. Intuitively, we define the complexity of a function in terms of its spectral content (or PSD), i.e., how fast the power of the function decays with frequency. Best-case functions (as defined later) are functions of low complexity (band limited or fast decaying). On the other hand, worst-case functions are high complexity functions with slowly decaying spectra.

**Best-Case Generalization Error.** We define our best-case function directly in the spectral domain with the radial mean power spectrum profile $\hat{\mathcal{P}}_l(\rho)$ which is a constant $c_l$ for $\rho < \rho_0$, and zero elsewhere. The constant $c_l$ indicates that the power spectrum is bounded. The best case gap can be thus obtained from Eq. 9 as follows:

**Proposition 4.** *The best-case generalization gap for isotropic homogeneous sample designs is*

$$gen(h) = \frac{\mu(\mathcal{S}^{d-1})}{N}c_l \int_0^{\rho_0} \rho^{d-1}\mathbb{E}(\hat{\mathcal{P}}_S(\rho))d\rho. \tag{10}$$

*Proof.* These results can be derived by plugging in best-case $\hat{\mathcal{P}}_l(\rho)$ in Eq. 9. $\square$

**Worst-Case Generalization Gap.** For the worst-case, we consider a function whose radial mean power spectrum $\hat{\mathcal{P}}_l(\rho)$ is a constant $c_l$ for $\rho < \rho_0$, and $c_l'\rho^{-d-1}$ elsewhere, where $c_l$ and $c_l'$ are non-zero positive constants. This spectral profile has a decay rate $O(\rho^{-d-1})$ for $\rho > \rho_0$.

**Proposition 5.** *The worst-case generalization gap for isotropic homogeneous sample designs is*

$$gen(h) = \frac{\mu(\mathcal{S}^{d-1})}{N}c_l \int_0^{\rho_0} \rho^{d-1}\mathbb{E}(\hat{\mathcal{P}}_S(\rho))d\rho + \frac{\mu(\mathcal{S}^{d-1})}{N}c_l' \int_{\rho_0}^\infty \rho^{-2}\mathbb{E}(\hat{\mathcal{P}}_S(\rho))d\rho. \tag{11}$$

*Proof.* These results can be derived by plugging in the worst-case $\hat{\mathcal{P}}_l(\rho)$ in Eq. 9. $\square$

Propositions 4 and 5 enable us to calculate the generalization gap of any isotropic sampling pattern as a function of the shape of sampling power spectra. Further, when an upper-bound on the power spectra of the loss function is known, one can deduce the corresponding error convergence rates.

# 7 Sampler-Specific Generalization Error Results

In the previous section, we obtained the best and worst-case generalization gap as a function of the sampling spectra. Next, we study the effects of different sample designs on the generalization gap.

**Random (or Poisson) Sampler**: A random sampler has a constant power spectrum since point samples are uncorrelated, i.e., $\mathbb{E}(\hat{\mathcal{P}}_S(\rho)) = 1, \forall \rho$.

**Proposition 6.** *For a random sampler, the best-case and the worst-case generalization gap can be obtained as:*

$$gen_b(h) = \mu c_l \rho_0^d / Nd, \text{ and } gen_w(h) = gen_b(h) + \mu c_l' \rho_0^{-1}/N.$$

*Proof.* These results can be derived by plugging in $\mathbb{E}(\hat{\mathcal{P}}_S(\rho)) = 1, \forall \rho$ in Eqns. 10 and 11. $\square$

**Blue Noise Sampler**: Blue noise design is aimed at replacing visible aliasing artifacts with incoherent noise, and their properties are typically defined in the spectral domain. We consider the step blue noise design defined as follows: (a) the spectrum should be close to zero for low frequencies, which indicates the range of frequencies that can be recovered exactly; (b) the spectrum should be a constant one for high frequencies, i.e. represent uniform white noise, which reduces the risk of aliasing. The low frequency band with minimal energy is referred to as the *zero region*. Formally,

$$P_S(\rho - \rho_z) = \begin{cases} 0 & \text{if } \rho \leq \rho_z, \\ 1 & \text{if } \rho > \rho_z. \end{cases} \tag{12}$$

The zero region $0 \leq \rho \leq \rho_z$ indicates the range of frequencies that can be represented with no aliasing and the flat region $\rho > \rho_z$ guarantees that aliasing artifacts are mapped to broadband noise.

Next, we derive optimal blue noise sample design in high-dimensions.

**Lemma 1.** *The PCF of a Step blue noise design of size $N$ in $d$ dimensions, for a given zero region $\rho_z$ is given by*

$$G(r) = 1 - (\rho_z/r)^{\frac{d}{2}} J_{d/2}(2\pi\rho_z r)/N, \tag{13}$$

*where $J_{d/2}(.)$ is the Bessel function of order $d/2$.*

*Proof.* These results can be derived by plugging in equation 12 in Theorem 1. $\square$

Lemma 1 helps us determine the maximum achievable zero region $\rho_z$, that does not violate realizability conditions.

**Lemma 2.** *The maximum achievable zero region using $N$ blue noise samples in $d$ dimensions is equal to inverse of the $d$-th root of the volume of a $d$-dimensional hyper-sphere with radius $1/\sqrt[d]{N}$,*

$$\rho_z^* = \sqrt[d]{N\Gamma(1 + d/2)/\pi^{d/2}},$$

*where $\Gamma(.)$ is the gamma function. Equivalently, we can determine the minimum number of samples needed to construct a step blue noise pattern, $N = \pi^{d/2}\rho_z^d/\Gamma(1 + d/2)$.*

**Proposition 7.** *For a blue noise design, the best-case and the worst-case generalization gap can be obtained as:*

$$gen_b(h) = \begin{cases} 0, & \text{if } \rho_0 \leq \rho_z^* \\ gen_b^{random}(h) - \mu c_l \Gamma(1 + d/2)/d\pi^{d/2}, & \text{otherwise} \end{cases}$$

$$gen_w(h) = \begin{cases} \mu c_l' \rho_z^{*-1}/N, & \text{if } \rho_0 \leq \rho_z^* \\ gen_w^{random}(h) - \mu c_l \Gamma(1 + d/2)/d\pi^{d/2}, & \text{otherwise}. \end{cases}$$

**Poisson Disk Sampler**: Without any prior knowledge of the function $\mathcal{F}$ of interest, a reasonable objective for sampling is that the samples should be random to provide an equal chance of finding features of interest. However, to avoid sampling only parts of the parameter space, a second objective is required to cover the space in $\mathcal{D}$ uniformly. Poisson Disk Sampling (PDS) is designed to achieve these objectives. In particular, the *step PCF* sampling pattern is a set of samples that are distributed

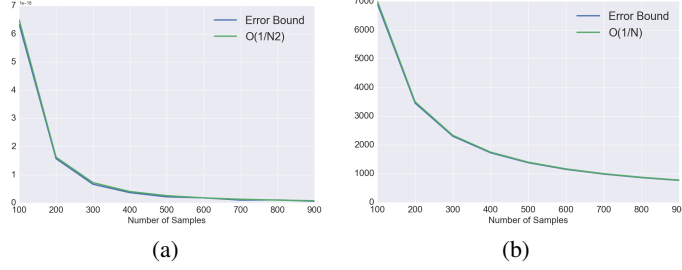

Figure 1: PDS convergence rate ($d = 2, c_l = 10^8, c'_l = 1.1, \rho_0 = 10^{-4}$) (a) Best-case, (b) Worst-case.

according to a uniform probability distribution (Objective 1: Randomness) but no two samples are closer than a minimum distance $r_{min}$ (Objective 2: Coverage). Formally,

$$G_S(r - r_{min}) = \begin{cases} 0 & \text{if } r \leq r_{min}, \\ 1 & \text{if } r > r_{min}. \end{cases} \tag{14}$$

Next, we derive optimal PDS sample design in high-dimensions.

**Lemma 8** ([18]). *The power spectra of a PDS design of size $N$ in $d$ dimensions, for a given $r_{min}$ is given by*

$$P_S(\rho - r_{min}) = 1 - N \left(2\pi r_{min}/\rho\right)^{d/2} J_{d/2}(\rho r_{min}), \tag{15}$$

*where $J_{d/2}(.)$ is the Bessel function of order $d/2$.*

Similar to the previous case, we can determining the maximum achievable $r_{min}$, that does not violate realizability conditions, for a given sample budget $N$.

**Lemma 9.** *The maximum achievable $r_{min}$ using $N$ PDS samples in $d$ dimensions is equal to inverse of $d$-th root of volume of a $d$-dimensional hyper-sphere with radius $\sqrt[d]{N}$,*

$$r_{min}^* = \sqrt[d]{\Gamma\left(1 + d/2\right)/\pi^{d/2}N},$$

*where $\Gamma(.)$ is the gamma function. Equivalently, we can also determine the minimum $N$ required to achieve a given $r_{min}$, $N = \Gamma(1 + d/2)/\pi^{d/2}r_{min}^d$.*

*Proof.* Proof is similar to Lemma 2. $\qquad\qquad\qquad\qquad\qquad\qquad\qquad\qquad\qquad\qquad\qquad\square$

**Proposition 10.** *For a PDS design, the best-case and the worst-case generalization gap can be obtained as:*

$$\begin{aligned} gen_b(h) &= gen_b^{random}(h) - \mu c_l(2\pi)^{\frac{d}{2}} r_{min}^* \int_0^{\rho_0} (\rho r_{min}^*)^{\frac{d}{2}-1} J_{d/2}(\rho r_{min}^*) d\rho, \\ gen_w(h) &= gen_b(h) + \frac{\mu c'_l \rho_0^{-1}}{N} - \mu c'_l (2\pi)^{\frac{d}{2}} r_{min}^{*d+2} \int_{\rho_0}^{\infty} (\rho r_{min}^*)^{-\frac{d}{2}-2} J_{\frac{d}{2}}(\rho r_{min}^*) d\rho. \end{aligned}$$

These integrals are complicated to compute and it is non-trivial to get closed-form bounds. Simplifications under simplistic assumptions are provided in the supplementary material. Propositions 6, 7 and 10 show that the shape of the power spectra has a major impact on the generalization gap–designs with optimized spectral properties are superior as compared to random designs.

# 8 Convergence Analysis

Next, we analyze the convergence of the generalization gap for different sample designs. This analysis will shed light into design principles for constructing optimal sample design.

## 8.1 Analysis with Sample Size

For random design, both the best and the worst-case generalization gaps converge as $O(1/N)$. For blue noise design, if best-case functions are bandwidth-limited with $\rho_0 \leq \rho_z^*$, then it can be perfectly recovered. However, when $\rho_0 > \rho_z^*$, the convergence is at the rate $O(1/N)$, which is the same as random design. For worst case functions, the error converges as $O(1/N\sqrt[d]{N})$ when $\rho_0 \leq \rho_z^*$ and as $O(1/N)$ when $\rho_0 > \rho_z^*$. This provides a theoretical justification for designing blue noise sample design with large zero-region $\rho_z$ for better performance. Note that, the convergence rate analysis of PDS design is not straightforward due to the involvement of Bessel functions under the integral in Proposition 10. Hence, we numerically analyze the convergence for PDS design. As showed in Figure 1, we observe that the best case convergence rate approximately behaves as $O(1/N\sqrt[d]{N}^b)$ with $b \geq 1$ and the worst case convergence behaves as $O(1/N)$.

## 8.2 Some Guidelines for Sample Design and Extensions

The main conclusion from our analysis is that samples with optimized spectral properties result in models with superior generalization. This conclusion is also corroborated via experiments in the next section. Our analysis shows that an ideal sample design power spectrum must approach to zero power as the frequency tends to zero (see Propositions 4 and 5). Power spectrum without oscillations in zero region achieves faster convergence rate as compared to power spectrum with oscillations. Ideally, one should target to generate sample designs whose power spectra has a large zero region $\rho_z$. However, the realizability conditions severely limit the range of realizable power spectra and hence in practice, this results in sample designs with very small $\rho_z$. A worthwhile direction for future work is to investigate sample designs with large zero regions. The proposed approach can also be used to study the effect of other state-of-the-art sample designs on the generalization gap. In many practical scenarios, it is possible to use information acquired from previous observations to improve the sampling process. As more samples are obtained, one can learn how to improve the sampling process by deciding where to sample next. These sampling feedback techniques are known as adaptive sampling. Our analysis provides a novel way to quantify the value of a sample in terms of the generalization gap. Another natural extension of our results is towards building importance sampling techniques, guided by spectral properties.

# 9 Experiments

Now, we corroborate our theoretical findings via experiments. We compare the generalization performance of different sample designs for regression.

**Experimental Setup.** In our experiments, we vary training sample set size from 200 to 1000. To generate blue noise and PDS designs, we use gradient descent based PCF matching approach as proposed in [21]. We use the implementation provided by the authors[3]. For both the experiments, we use neural network with two hidden layers, with 200 and 100 nodes respectively, each followed by a LeakyReLu activation function. For training algorithm, we use ADAM optimizer with learning rate and batch size to be 0.01 and 64, respectively. We evaluate the generalization performance of neural networks learnt using different sample designs based on root mean square error (RMSE) on $10^3$ unseen regular grid based test samples. All the results are averaged over 20 independent realizations.

**Synthetic Functions.** In this experiment, we consider regression problem of learning analytical functions and perform a comparative study of different sample designs, in terms of their generalization performance. We consider two synthetic functions with known but different spectral behavior: a) disk function: $y = 5$ if $|x| < 6$ ($y = 0$ otherwise), and b) exponential function: $y = 10 * exp(-30 * (|x|^2))$ where $x \in [0, 1]^3$. In Figure 2, we show radial average of both the functions and their power spectral densities. Note that exponential function is not bandwidth-limited but smooth enough to have an exponential decay rate for its PSD. On the other hand, spectral profile of the disk function has a decay rate of $O(\rho^{d-1})$ as assumed in Section 6. For both of the functions, we see that models trained on blue noise and PDS sample designs generalize significantly better for all sampling budgets as compared to models trained on random sample design.

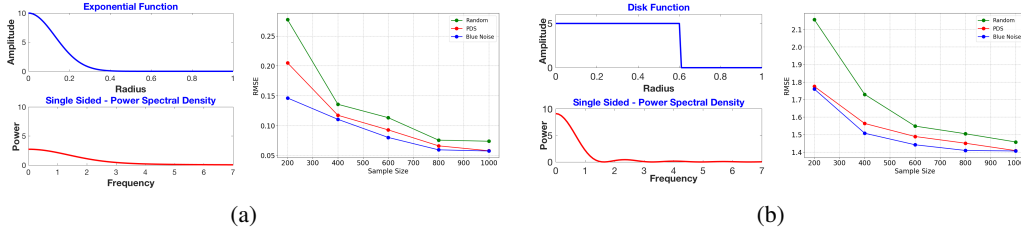

Figure 2: Generalization comparison on synthetic functions: (a) exponential, (b) disk.

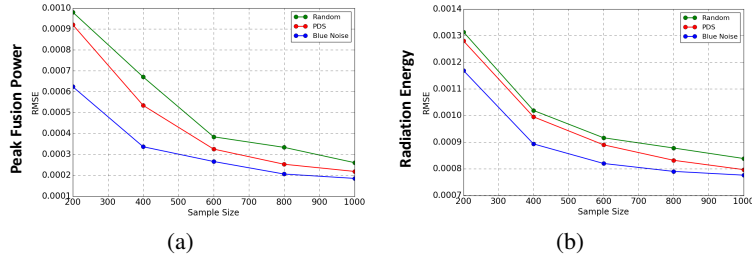

Figure 3: Generalization comparison on ICF application: (a) peak fusion power, (b) radiation energy.

**Inertial Confinement Fusion (ICF) Simulator.** Next, we consider a scientific machine learning problem of learning a regression model for an inertial confinement fusion (ICF) simulator developed at the National Ignition Facility (NIF). The NIF is aimed at demonstrating inertial confinement fusion (ICF), that is, thermonuclear ignition and energy gain in a laboratory setting. We use the NIF JAG simulator [4] with different input parameters, such as, laser power, pulse shape etc. For each simulation run, several output quantities, such as peak fusion power, yield, etc., are obtained. In this experiment, we vary three input parameters and study the problem of learning a model to regress peak fusion power and radiation energy. Note that, the function and its spectral behavior is not known in this experiment and its may not comply with any of our assumptions. In Figure 3, we observe that regression error patterns are consistent with our observations in the previous experiment. Blue noise design performs the best followed by the PDS design. This shows that our finding that spectral designs are superior compared to random designs hold even in this real-world setting.

The performance gain with both synthetic function and ICF simulator can be credited to superior spectral properties of blue noise and PDS designs compared to random designs. These observations corroborate our theoretical results which show that the shape of the power spectra has a major impact on the generalization gap and sampling designs with optimized spectral properties (i.e., blue noise and PDS) are superior to random designs. Further, the gain of spectral designs is higher in low-sampling regime which makes spectral designs an attractive solution in small-data ML applications.

## 10 Conclusions

We presented a statistical mechanics framework to study effect of task-agnostics sample designs on the generalization gap of ML models. We showed that the generalization gap is related to the power spectra of a sample design and the function of interest. We also analyzed the generalization gap of two state-of-the-art space-filling sample designs, and quantified their gain over random design in a closed-form. Finally, we provided design guidelines towards constructing optimal sample designs for a given problem. There are still many interesting questions that remain to be explored such as an analysis of the generalization gap for cases where input domain is a non-linear manifold. Analysis with a specific loss function can also be pursued. Other potentially worthwhile directions are designing significantly higher quality sample designs than currently possible, adaptive sampling, and importance sampling.

## Acknowledgments

This work was supported by the U.S. Department of Energy by Lawrence Livermore National Laboratory under Contract DE-AC52-07NA27344.

## Broader Impact

In this paper, we introduce a statistical mechanics framework to understand the effect of sample design on the generalization gap of ML algorithms. Our framework could be applied to a wide range of applications, including scientific ML, design and optimization in engineering, agricultural experiments, and many more. It can also play an important building block for several ML problems, such as, supervised ML, neural network training, image reconstruction, reinforcement learning, etc. We expect that our framework will significantly improve the quality of inference and our current understanding in several science and engineering applications where ML is applied. Our focus on this paper has been understanding the effect of sample design on the generalization gap, however, in several applications we may additionally want to understand implication of a sample design on fairness, robustness, privacy, etc. This is an unexplored area in sample design and we encourage researchers to understand and mitigate the risks arising from task-agnostic designs in these contexts.

## Footnotes

[1]Whether or not these two conditions are not only necessary but also sufficient is still an open question (however, no counterexamples are known).

[2]This can further be extended to more complex hypothesis dependent analysis, e.g., by applying Hoeffding's inequality to each fixed hypothesis to obtain uniform bounds.

[3]https://github.com/gowthamasu/Coveragebasedsampledesign

[4] https://github.com/rushilanirudh/macc

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
