[Supplementary Material]

# A  Description of Sample Designs

In this paper, we consider three different families of sample design for our generalization gap analysis, namely random, blue noise and Poisson disk designs. Figure 4 illustrates the sample design along with their spectral/spatial properties for $d = 2$ and $N = 1000$. Note that, we show the 2D PSD here, though our analysis assumes isotropic distributions and hence uses radially averaged 1D-PSD.

Figure 4: Sample design along with their spatial/spectral properties considered in our analysis.

# B  Proof of Theorem 1 from the main paper

We know that the PSD and PCF of a point distribution are related via the Fourier transform as follows:

$$
\begin{aligned}
P(\mathbf{k}) &= 1 + \rho F\left(G(\mathbf{r}) - 1\right) \\
&= 1 + N \int_{\mathbb{R}^d} \left(G(\mathbf{r}) - 1\right) \exp(-2\pi i \mathbf{k}.\mathbf{r}) d\mathbf{r}
\end{aligned}
$$

where $F(.)$ denotes the $d$-dimensional Fourier transform. Using symmetry of the Fourier transform, we have

$$
G(\mathbf{r}) = 1 + \frac{1}{N} F\left(P(\mathbf{k}) - 1\right).
$$

Next, we use polar coordinates with the $z$ axis along $\mathbf{k}$, so that $\mathbf{k}.\mathbf{r} = \rho r \cos\theta$ where $\rho = |\mathbf{k}|$ and $r = |\mathbf{r}|$. For radially symmetric PCF, we have $G(\mathbf{r}) = G(r)$ and the above relationship can be rewritten as

$$
\begin{aligned}
G(r) = 1 + \frac{1}{N} \int_0^\infty \int_0^\pi \exp\left(-2\pi i \rho r \cos(\theta)\right) \\
(P(\rho) - 1)\omega \sin(\theta)^{d-2} d\theta r^{d-1} \, d\rho
\end{aligned}
$$

where $\omega$ is the area of unit sphere in $(d-1)$ dimension. Next, using the identity involving bessel function of order $v$, i.e.,

$$
\begin{aligned}
J_v(2\pi t) = \frac{(2\pi t)^v}{(2\pi)^{v+1}} \int_0^\pi \exp\left(-2\pi i t \cos(\theta)\right) \\
\omega \sin(\theta)^{2v} \, d\theta,
\end{aligned}
$$

we obtain

$$G(r) = $$
$$1 + \frac{2\pi}{N} r^{1-\frac{d}{2}} \int_0^\infty \rho^{\frac{d}{2}-1} J_{\frac{d}{2}-1}(2\pi\rho r)\rho(P(\rho)-1) \, dk.$$

## C  Proof for Theorem 2 from the main paper

The proof follows from [23] and provided here for completeness.

Let us denote the Fourier domain without the DC peak frequency as $\Theta$. Since homogeneous sampling patterns have statistical properties that are invariant to translation, it is equivalent to studying the error due to the translated version of each realization, with the average computed over all translations. Formally, we can treat the torus as the group of translations, so that $\tau(S)$ denotes the translation of $S$ by an element $\tau \in \mathcal{T}^d$. Then, averaging equation (7) over all translations of $S$, we get:

$$\text{gen}(h) \triangleq \frac{1}{N^2} \int_{\mathcal{T}^d \times \Theta \times \Theta} \mathbb{E}(\mathcal{F}_{\mathcal{T}(S),l}(\mathbf{k}, \mathbf{k}')) d\mathbf{k} d\mathbf{k}' d\tau,$$

$$= \frac{1}{N^2} \int_{\mathcal{T}^d \times \Theta \times \Theta} \mathbb{E}(\mathcal{F}_{S,l}(\mathbf{k}, \mathbf{k}')) \times$$
$$\exp^{i2\pi\tau \cdot (\mathbf{k}'-\mathbf{k})} d\mathbf{k} d\mathbf{k}' d\tau, \tag{16}$$

where the exponential arises from the translation of the sample design by a vector $\tau$ in the Fourier domain. When $\mathbf{k} \neq \mathbf{k}'$, the integral of the exponential part equals zero, so that only the case $\mathbf{k} = \mathbf{k}'$ contributes to the variance. Hence, we can remove one integral over $\Theta$ and obtain

$$\text{gen}(h) \triangleq \frac{1}{N^2} \int_\Theta \mathbb{E}(\mathcal{F}_{S,l}(\mathbf{k}, \mathbf{k})) \int_{\mathcal{T}^d} d\mathbf{k} d\tau \tag{17}$$

$$= \frac{1}{N^2} \int_\Theta \mathbb{E}(\|\mathcal{F}_{S,l}(\mathbf{k}, \mathbf{k})\|^2) d\mathbf{k} \tag{18}$$

Finally, denoting the power spectrum of the loss by $\mathcal{P}_l$ and the power spectrum of the sample design normalized by $N$ as $\mathcal{P}_S$, and leveraging the fact that $\|\mathcal{F}_{S,l}(\mathbf{k}, \mathbf{k})\|^2 = \|\mathcal{F}_S(\mathbf{k})\|^2 \cdot \|\mathcal{F}_l(\mathbf{k})\|^2$,

$$\text{gen}(h) \triangleq \frac{1}{N} \int_\Theta \mathbb{E}(\mathcal{P}_S(\mathbf{k}))\mathcal{P}_l(\mathbf{k}) d\mathbf{k} \tag{19}$$

This provides the expression for the generalization gap in terms of the power spectra of both the sampling pattern and the loss function in the toroidal domain.

## D  Proof of Lemma 2 from the main paper

Note that, for a Step blue noise configuration to be realizable, it is sufficient to show that the corresponding PCF is non-negative. Thus, we have

$$G(r) \geq 0$$

$$\Leftrightarrow \quad 1 \geq \frac{1}{N} \left(\frac{\rho_z}{r}\right)^{\frac{d}{2}} J_{\frac{d}{2}}(2\pi\rho_z r)$$

$$\Leftrightarrow \quad 1 \geq \frac{1}{N} (\rho_z\sqrt{2\pi})^d \frac{J_{\frac{d}{2}}(2\pi\rho_z r)}{(2\pi\rho_z r)^{\frac{d}{2}}}$$

$$\Leftrightarrow \quad 1 \geq \frac{1}{N} \frac{(\rho_z\sqrt{2\pi})^d}{2^{\frac{d}{2}}\Gamma\left(1+\frac{d}{2}\right)}$$

$$\Leftrightarrow \quad \rho_z \leq \left(\frac{N\Gamma\left(1+\frac{d}{2}\right)}{\pi^{d/2}}\right)^{1/d}.$$

In the last inequality, we have used the following approximation

$$J_v(x) = \frac{(x/2)^v}{\Gamma(1+v)}.$$

## E   Proof of Proposition 7 from the main paper

The best-case generalization gap for blue noise design is given by:

$$\text{gen}_b(h) \;=\; \frac{\mu(\mathcal{S}^{d-1})}{N}c_l\int_0^{\rho_0}\rho^{d-1}P_S(\rho-\rho_z^*)d\rho. \tag{20}$$

Note that, when $\rho_0 \leq \rho_z^*$ the best-case generalization error $\text{gen}_b(h) = 0$, and when $\rho_0 > \rho_z^*$, we have

$$
\begin{aligned}
\text{gen}_b(h) &= \frac{\mu(\mathcal{S}^{d-1})}{N}c_l\int_{\rho_z^*}^{\rho_0}\rho^{d-1}d\rho = \frac{\mu c_l}{N}\left[\frac{\rho_0^d - \rho_z^{*d}}{d}\right]\\
&= \text{gen}_b^{\text{random}}(h) - \frac{\mu c_l \rho_z^{*d}}{Nd}\\
&= \text{gen}_b^{\text{random}}(h) - \frac{\mu c_l \Gamma\left(1+d/2\right)}{d\pi^{d/2}}
\end{aligned}
\tag{21}
$$

The worst-case generalization gap can be obtained as:

$$
\begin{aligned}
\text{gen}_w(h) &= \frac{\mu(\mathcal{S}^{d-1})}{N}c_l\int_0^{\rho_0}\rho^{d-1}P_S(\rho-\rho_z^*)d\rho\\
&+ \frac{\mu(\mathcal{S}^{d-1})}{N}c_l'\int_{\rho_0}^{\infty}\rho^{-2}P_S(\rho-\rho_z^*)d\rho
\end{aligned}
\tag{22}
$$

Note that, when $\rho_0 \leq \rho_z^*$ the worst-case generalization error $\text{gen}_w(h) = \dfrac{\mu c_l'}{N\rho_z^*}$, and when $\rho_0 > \rho_z^*$,

$$
\begin{aligned}
\text{gen}_w(h) &= \frac{\mu(\mathcal{S}^{d-1})}{N}c_l\int_{\rho_z^*}^{\rho_0}\rho^{d-1}d\rho\\
&+\frac{\mu(\mathcal{S}^{d-1})}{N}c_l'\int_{\rho_0}^{\infty}\rho^{-2}d\rho\\
&= \frac{\mu c_l}{N}\left[\frac{\rho_0^d - \rho_z^{*d}}{d}\right] + \frac{\mu c_l' \rho_0^{-1}}{N}\\
&= \text{gen}_b(h) + \frac{\mu c_l' \rho_0^{-1}}{N}\\
&= \text{gen}_w^{\text{random}}(h) - \frac{\mu c_l \Gamma\left(1+d/2\right)}{d\pi^{d/2}}
\end{aligned}
\tag{23}
$$

## F   Proof of Proposition 10 from the main paper

For PDS design, the best-case generalization gap can be obtained as:

$$
\begin{aligned}
&\text{gen}_b(h)\\
&= \frac{\mu(\mathcal{S}^{d-1})}{N}c_l\int_0^{\rho_0}\rho^{d-1}P_S(\rho-r_{min}^*)d\rho\\
&= \frac{\mu c_l \rho_0^d}{Nd} - \mu c_l(2\pi r_{min}^*)^{d/2}\int_0^{\rho_0}\rho^{\frac{d}{2}-1}J_{d/2}(\rho r_{min}^*)d\rho\\
&= \text{gen}_b^{\text{random}}(h) - \mu c_l(2\pi r_{min}^*)^{d/2}\int_0^{\rho_0}\rho^{\frac{d}{2}-1}J_{d/2}(\rho r_{min}^*)d\rho\\
&= \text{gen}_b^{\text{random}}(h)\\
&\quad -\mu c_l(2\pi)^{d/2}r_{min}^*\int_0^{\rho_0}(\rho r_{min}^*)^{\frac{d}{2}-1}J_{d/2}(\rho r_{min}^*)d\rho
\end{aligned}
\tag{24}
$$

Figure 5: (a) Maximum achievable $\rho_z$ with $N = 10$ samples of blue noise design at varying dimensions $d$, (b) Minimum number of sampling points needed to achieve $\rho_z = 5$ at varying dimensions $d$, (c) Maximum achievable $r_{min}$ with $N = 10$ samples of PDS at varying dimensions $d$, (d) Minimum number of sampling points needed to achieve $r_{min} = 1$ at varying dimensions $d$.

The worst-case generalization gap can be obtained as:

$$
\begin{aligned}
& \mathrm{gen}_w(h) \\
=\ & \mathrm{gen}_b(h) + \frac{\mu(\mathcal{S}^{d-1})}{N}c_l' \int_{\rho_0}^{\infty} \rho^{-2} P_S(\rho - r_{min}^*)d\rho \\
=\ & \mathrm{gen}_b(h) + \frac{\mu}{N}c_l' \int_{\rho_0}^{\infty} \rho^{-2} d\rho \\
& -\mu c_l'(2\pi r_{min}^*)^{d/2} \int_{\rho_0}^{\infty} \rho^{-\frac{d}{2}-2} J_{d/2}(\rho r_{min}^*)d\rho \\
=\ & \mathrm{gen}_b(h) + \frac{\mu c_l' \rho_0^{-1}}{N} \\
& -\mu c_l'(2\pi r_{min}^*)^{d/2} \int_{\rho_0}^{\infty} \rho^{-\frac{d}{2}-2} J_{d/2}(\rho r_{min}^*)d\rho \\
=\ & \mathrm{gen}_b(h) + \frac{\mu c_l' \rho_0^{-1}}{N} \\
& -\ \mu c_l'(2\pi)^{\frac{d}{2}} r_{min}^{*d+2} \int_{\rho_0}^{\infty} (\rho r_{min}^*)^{-\frac{d}{2}-2} J_{\frac{d}{2}}(\rho r_{min}^*)d\rho
\end{aligned}
\tag{25}
$$

## G  Generalization Gap Bounds for Poisson Disk Sample Design

**Best Case**

$$
\begin{aligned}
\mathrm{gen}_b(h) =\ & \mathrm{gen}_b^{\mathrm{random}}(h) \\
& - \mu c_l(2\pi)^{d/2} r_{min}^* \int_0^{\rho_0} (\rho r_{min}^*)^{\frac{d}{2}-1} J_{\frac{d}{2}}(\rho r_{min}^*)d\rho
\end{aligned}
\tag{26}
$$

$$
\begin{aligned}
=\ & \mathrm{gen}_b^{\mathrm{random}}(h) \\
& - \mu c_l(2\pi)^{d/2} \int_0^{r_{min}^*\rho_0} \rho^{\frac{d}{2}-1} J_{d/2}(\rho)d\rho
\end{aligned}
\tag{27}
$$

$$
\begin{aligned}
\leq\ & \mathrm{gen}_b^{\mathrm{random}}(h) \\
& - \frac{\mu c_l(2\pi)^{d/2} 2^{\frac{-d}{2}} (\rho r_{min}^*)^d}{d\Gamma(1+\frac{d}{2})}\left(1 - \frac{1}{8}\frac{d(\rho r_{min}^*)^2}{(1+\frac{d}{2})^2}\right)
\end{aligned}
\tag{28}
$$

$$
=\ \frac{\mu c_l \Gamma^{\frac{2}{d}}(1+\frac{d}{2})\rho_0^{2+d}}{8\pi(1+\frac{d}{2})^2}\frac{1}{N^{1+\frac{2}{d}}}
\tag{29}
$$

The second inequality above is based on the series form of the hypergeometric function and the assumption that $N$ is a large number.

# H Convergence Analysis of Generalization Gap with Dimensions

In this section, we report some interesting observations when analyzing the generalization gap with increasing dimensions.

## H.1 Analysis with Traditional Metrics

We study the limiting behavior of $\rho_z^*$ and $r_{min}^*$ as $d$ approaches infinity. We show that the analysis with conventional metrics to characterize the zero region, i.e., the range of frequencies that can be represented with no aliasing, provides some rather counter-intuitive results.

**Proposition 11.** *As the dimension $d$ approaches infinity, the maximum achievable zero region for blue noise design, with a fixed $N$, goes to infinity, i.e., $\lim_{d\to\infty} \rho_z^* = \infty$ and, the minimum number of samples needed to achieve a zero region $\rho_z$ approaches zero, i.e., $\lim_{d\to\infty} N = 0$.*

Intuitively, with growing $d$, one might expect $\rho_z^* \to 0$ and $N \to \infty$. To better understand this result, we study the relationship between these two quantities and the volume of a hyper-sphere. One of the surprising facts about a sphere in high dimensions is that as the dimension increases, the volume of the sphere goes to zero which justifies the above results. Our intuitions about space are formed in two or three dimensions and often do not hold in high dimensions. A more surprising fact is that $\rho_z^*$ and $N$ are not monotonic functions with respect to $d$ (see Figure 5(a) and 5(b)). Either a steady increase or a steady decrease seems more plausible than having these two quantities grow for a while, then reach a peak at some finite value of $d$, and thereafter decline. This behavior has also been observed in high dimensional geometry while analyzing the volume of a hypersphere, however, no physical interpretation or intuition currently exists for this open research problem [12].

Similarly, we study the asymptotic behavior of the maximum achievable $r_{min}$ for a fixed sample budget, and equivalently the minimum number of samples required to achieve a PDS with a given $r_{min}$, as the dimension grows to infinity.

**Proposition 12.** *As the dimension $d$ approaches infinity, the maximum achievable $r_{min}$ for PDS design, with a fixed number of samples, goes to infinity, i.e., $\lim_{d\to\infty} r_{min}^* = \infty$ and, the minimum number of samples needed to achieve a $r_{min}$ also approaches infinity, i.e., $\lim_{d\to\infty} N = \infty$.*

The results in the proposition above are reasonable, since the space is growing exponentially fast.

## H.2 New Metrics for Reliable Analysis

Analysis with the metric $\rho_z^*$ which are based on the amplitude of the frequency vector, i.e., $\mathbf{k}$, to characterize the zero region, leads to inconsistent results in high dimensions. We argue that comparing $\rho_z^*$ and $r_{min}^*$ across different dimensions is not accurate, and these inconsistent results are a byproduct of the improper comparisons. Note that, each $d$-dimensional space is comprised of a different range of frequency components, and comparing the magnitude of the frequency vector directly across dimensions is questionable. In particular, for a valid comparison of volumes across dimensions, we propose to measure them in terms of a standard volume in that dimension, i.e., unit hypercube or the measure polytope, which has a volume of $1$ in all dimensions. Further, as the dimension $d$ increases, the maximum possible distance between two points in a hypercube grows as $\sqrt{d}$. Consequently, to have the same scale across dimensions, we normalize the radius of the hypersphere by a factor $\sqrt{d}$. In summary, we introduce the *relative zero region*, i.e., $\hat{\rho}_z^* = \rho_z^* / \sqrt{d}$ ($\hat{r}_{min}^* = r_{min}^* / \sqrt{d}$) for meaningful convergence analysis across dimensions.

**Proposition 13.** *As dimension $d$ approaches infinity, the maximum achievable relative $\rho_z^*$ converges to a constant, i.e.,*

$$\lim_{d\to\infty} \hat{\rho}_z^* = \frac{1}{\sqrt{2\pi e}},$$

*and the minimum number of blue noise samples needed to achieve $\hat{\rho}_z^*$ goes to infinity.*

*Proof.* To prove the first identity, note that $\frac{\rho_z^*}{\sqrt{d}} = \frac{\sqrt[d]{N}}{\sqrt{\pi d}} \sqrt[d]{\Gamma\left(1 + \frac{d}{2}\right)}$ and invoke Stirling's approximation, i.e., $\Gamma(1 + m) = \left(\frac{m}{e}\right)^m \sqrt{2\pi m}$. Now, the required result can be obtained by letting $d$ approach infinity. The second identity can be proved in a similar manner. $\qquad\square$

Similarly, we study the asymptotic behavior of $\hat{r}^*_{min}$ for PDS sampling.

**Proposition 14.** *As dimension $d$ approaches infinity, the maximum achievable relative $r^*_{min}$ converges to a constant, i.e.,*

$$\lim_{d\to\infty} \hat{r}^*_{min} = \frac{1}{\sqrt{2\pi e}}$$

*and, the minimum number of PDS samples needed to achieve $\hat{r}^*_{min}$ goes to infinity.*

*Proof.* Proof is similar to the one as in Proposition equation 13 and, thus, omitted. □

The results in Propositions 13 and 14 show interesting limiting behaviors of both blue noise and PDS designs.