[Reviews · NeurIPS 2020]

Review 1

Summary and Contributions: summary: in this paper, the authors describe the generalization gap in terms of the power spectrum of the input sample distribution. this allows the authors to estimate analytically the generalization gap for sampler designs that can be described in spectral terms. they evaluate the generalization gap for three sample design functions: random (poisson), poisson disk sampling, and blue noise. they show empirical results comparing these samplers on two synthetic function approximation tasks and ICF. contributions: (1) theoretical: spectral analysis of generalization gap (significance: medium) (2) methodological: advocate for task-agnostic sample designs that can be expressed in spectral terms and analytically compared in terms of generalization gap and convergence. (significance: medium) (3) empirical: empirical support for these samplers on two empirical tasks (significance: low, given the small number of experiments and

Strengths: the work is theoretically grounded, and addresses the important and relevant problem of optimizing data sampling in a task-agnostic way. it introduces a general theoretical formalism for describing the spectral gap. i

Weaknesses: not clear how these results generalize beyond the specific samplers explored. in practice, are the spectral properties of samplers often known? can the conclusions be applied to space-filling designs generally, as claimed? is it the spectral properties that are optimal, or are the spectral properties picking up something else? the authors argue that blue noise + PDS optimize "spectral properties" -- what are optimal spectral properties? seems potentially circular. experimental results are a bit sparse -- sample size only goes up to 1000, and only one architecture was explored. In Fig S3b, random catches up to PDS+blue noise: would it surpass for greater sample sizes? -- after rebuttal -- the rebuttal has clarified some of the things i hadn't understood about the contributions and edited a figure that i was particularly concerned about. i've raised my score accordingly. thanks for the clarifications.

Correctness: as far as i know yes.

Clarity: overall this paper was clearly written and easy to follow. minor: all experimental results are relegated to supplement, which seems unusual. some variables are confusingly introduced (e.g. G and r were poorly defined) and occasionally the point of introducing certain relationships was not clear (e.g. why was it to relate PSD and PCF?). some proof explanations were a bit sparse (e.g. props 4, 5).

Relation to Prior Work: as far as i know yes.

Reproducibility: Yes

Additional Feedback:


Review 2

Summary and Contributions: The authors present a new method for studying and comparing sample designs (i.e. schemes for sampling training points). The method relies on a novel relation between the generalization gap and power spectral density of the sample design. This relationship also requires the PSD of the loss function, however, which the authors circumvent by considering worst- and best- case loss functions. The best and worst generalization gaps are then calculated exactly for 3 sample designs. the convergence of these gaps suggest that the Blue Noise and Poisson Disc samplers should outperform a random sampler with uncorrelated samples. This is confirmed in experiments that are shown in the Supplementary.

Strengths: The paper is well written and clearly develops the main results. The connection between the spectral properties of a design and the generalization gap is novel and presents a promising way forward for analyzing designs. The assumptions and corresponding simplifications used to derive results for practically useful samplers are clearly stated and well-justified. Overall, this paper approaches an important problem with rigorous analysis that yields (some) practically realizable results; it is a good contribution.

Weaknesses: My main concern with this paper is the relatively small amount of criticism of the theory. There are a number of places where more thorough examination of both the assumptions and results would be helpful in understanding the advantages and limitations of the proposed method. For example, the authors’ ability to derive results for the 3 considered samplers relies on the definition of best and worst case loss functions. These are defined in terms of spectral properties, which makes it difficult to understand how they relate to loss functions to those used in practice. An analysis of this for simple model classes would help to understand the results of the paper. Additionally, it would be helpful to understand the precision that the theory is able to make predictions about potential sample designs. For example, the experiments in the supplement show that the theory correctly predicts that the random sampler perform worse than the other samplers, but is it able to distinguish between the PDS and Blue Noise? The authors note that their results provide “theoretical justification for designing blue noise sample design with large zero-region for better performance”, but does the theory allow one to optimize the size of the zero region? Probing these questions would strengthen the paper.

Correctness: The results appear to be correct.

Clarity: The paper is well written and clear.

Relation to Prior Work: How this work is novel and connected to previous work is clearly discussed. The only possible improvement to this would be a more thorough description of the differences between the spectral analysis in reference 17 and this paper beyond the fact that ref. 17 does not consider generalization gap.

Reproducibility: Yes

Additional Feedback: Minor typo: the question mark in line 21 should be a period.


Review 3

Summary and Contributions: This paper investigates the following question: assuming a supervised ML model where training data input are related to the output, or the response, through some unknown function to be learned, how does one most effectively collect training data input, assuming that one can control the data collection process? This is relevant when curated datasets are not available a priori and collecting the data is costly. In particular, they study the effect of training data sampling properties on the generalization gap for algorithms, where the generalization gap is defined as the expected difference between the population risk and the empirical risk. Their analysis uses a statistical mechanics framework that represents the generalization gap in terms of the power spectra of the sample and the function to be learned. Using this, they are able to show that space-filling sampling designs (that aim to cover the sample input space uniformly) outperform random designs in terms of generalization gap, and can characterize this discrepancy. -- after rebuttal -- I have read the rebuttal and am satisfied with the authors' comments wrt to my review. Thanks to the authors for their thoughtful response.

Strengths: One strength of this work is that most theoretical frameworks for studying generalization with respect to sampling design only address random i.i.d. designs that do not cover the generic sampling designs, like a space-filling design, studied here. In particular, they are able to characterize in closed form the generalization gains in using state-of-the-art space-filling designs, like Poisson disk sampling or blue noise, over such random design. The analysis also allows one to optimize the sampling design for specific ML problems in terms of generalization.

Weaknesses: The only weaknesses I can see are that the paper could do a better job of differentiating tis results from those of [17] and also that it could be written a bit more clearly (as is elaborated below in the ‘Clarity’ section).

Correctness: The claims appear correct to this reviewer and the empirical methodology is well justified.

Clarity: I believe that the writing in the paper could be improved. It is my impression that the authors wanted to fit a lot of results into their 8 pages and this meant that, at times, the paper read as if it is just a list of theoretical results. This is especially true in Sections 5-7 and I think it would be easier to understand these results if there was some additional discussion. I found the use of PSD and PDS to mean, respectively, power spectral density and Poisson disk sampling to be confusing.

Relation to Prior Work: It appears to this reviewer that the authors include an appropriate bibliography. It may be helpful to further clarify differences from the work in [17]. Specifically, it is said that [17] proposes using a spectral framework (like the one used in this work) to quantify the space-filling properties of such sampling designs and to demonstrate superiority over other designs, however [17] does not rigorously characterize generalization performance. So my question is if [17] does not characterize generalization, how exactly do they demonstrate superiority of space-filling over other designs?

Reproducibility: Yes

Additional Feedback: Some additional comments: -- Could you provide some more justification for the best-case and worst-case loss function power spectra used in Propositions 4 and 5. Is it obvious that these choices are in fact the best and worst cases? Some suggestions/typos: -- The idea of isotropic designs is explained on page 5 where the authors say that this means the power spectrum is radially symmetric but the term ‘isotropic’ design is used interchangeably with the term ‘radially symmetric’ design as soon as page 3. -- In the comment on realizability it is probably worth defining what realizability means and also noting that the result stated is only for isotropic designs. -- It would be helpful to define what is meant by ‘homogenous’ sample designs in Theorem 2. -- The spacing is off in the equation in Proposition 6.

[Author Response · NeurIPS 2020]

We thank the reviewers for their insightful and positive feedback. We are encouraged that they found our idea to be
important (R3 , R4 , R7 ), theoretically grounded (R3 , R4 , R7 ), clear (R3 , R4 ), and novel (R4 , R7 ); and our analysis
to be rigorous and insightful (R4 , R7 ). We are glad that both R4 and R7 recognized the importance and novelty of this
work and suggested an acceptance. Though R3 has recommended a rejection, the weaknesses identified by R3 are in
fact clarifying questions. Very interestingly, all the pointed weaknesses/questions are in fact the biggest strengths of our
framework. Hence, we are hopeful that R3 will champion our paper for acceptance.

**R3 Contributions and Significance**: Currently, there do not exist any theoretical framework that allows the study of
generalization of generic task-agnostic sample designs (e.g., space-filling). Our work proposes a fundamentally novel
approach to this problem and significantly advances our understanding of sample design in the context of ML. This was
achieved by connecting ideas from disparate fields – spectral analysis of sampling design (from DOE) and generalization
error (from ML) using a statistical mechanics framework. In this process, we have made new contributions to both
fields. In ML, our analysis enabled us go beyond the generalization analysis of random i.i.d. designs or other simple
probabilistic variants. In DOE, it provided the much needed theoretical underpinning for the apparent superiority of
space-filling designs over random designs. Further, this is the first work to construct optimal blue noise designs in
dimensions $d > 2$. Finally, we derived new results for high-dim. PDS and blue noise (see supp mat). We hope this
addresses R3 's concerns regarding the significance of our contributions.

**R3 It is not clear how these results generalize beyond the specific samplers explored.** Unlike existing approaches,
our framework supports generic sample designs and this is **precisely our major contribution**. By combining Theorem
2 with Eq. 5, one can calculate the generalization gap of arbitrary sample designs (see line 156). We continued making
our analysis more focused in rest of the paper by deriving results for SOTA samplers (as pointed our by R4 and R7 ).
One can use our tools to study other complex designs (e.g., Latin Hypercube, Quasi Monte Carlo) that is currently not
possible with traditional approaches.

**R3 In practice, are the spectral properties of samplers often known?** Spectral properties (i.e., power spectra) of
any sample design can be computed (line 94 and Figure 1 in supp mat). While in some cases, their analytical forms (or
upper bounds) are known – our theory is not limited to analytical PSDs, thus making it **broadly applicable**.

**R3 Is it the spectral properties that are optimal, or they are picking up something else?** We are not sure if we
understand this question clearly. Every sample design can be represented in its spectral form and our analysis suggests
that for an ideal design, the power spectrum must approach to zero power at low freq. (line 266). As these claims are
proved rigorously, there are no confounding variables as it might have been the case with only empirical results.

**R3 Blue noise optimize spectral properties-what are optimal spectral properties? seems potentially circular.**
This result is not circular in any way. Our guidelines for optimal power spectra form (line 265) are derived for any
generic sample design (Proposition 3 and Theorem 2) and it so happens that blue noise and PDS satisfy them.

**R3 Experiments are bit sparse.** In addition to neural network regression on synthetic
functions and real-world scientific simulator in the paper, we further experimented with
(Randomforests, GBT), and results were consistent (will add to revision). Result that
random design matches PDS (Fig S3b) was an artifact and is removed by increasing
the number of independent realizations from 20 to 30. Due to theoretical nature of this
work, we hope the lack of large scale experiments will not be considered as a weakness.

**R3 Certain relationships not clear (why relate PSD and PCF).** Theorem 1 along
with realizability conditions establish a fundamental relationship between the PSD and PCF of designs. This relationship
is required for constructing optimal forms of PDS and blue noise, and carrying out analysis only on realizable spectras
(not every power spectra has corresponding sample design, though, other way around is always true) (see line 111).

**R3 Some proof explanations were a bit sparse (e.g. props 4, 5).** These results can be obtained simply by plugging
in best- and worst-case PSD (as defined in the paper) in Eq 9. We will add this additional step in the revised version.

**R4 , R7 Comparison with [17].** The only overlap between both papers is the formal construction of PDS, which is
needed for Lemma 8 of our paper. While [17] and other space-filling papers design PDS heuristically/empirically, our
analysis is entirely different and theoretically grounded.

**R4 . Limitation of theory** (or potential extensions) are (e.g., probabilistic bounds, manifold sampling, adaptive
sampling). Theory allows one to optimize the zero region (see Lemma2). Loss specific analysis and distinguishing PDS
and Blue Noise empirically are excellent suggestions and will be added as the future work in the revised version.

**R7 . Technical reason behind best and worst case choices** is given in [6], and will be clarified in the revised version.
The definition of isotropic and homogeneous designs will also be clarified. Sorry for the typo – the realizability
condition will be written in its generic form instead of the radial form.

[Meta-Review · NeurIPS 2020]

The authors present a new method based on statistical mechanics for studying and comparing sample designs. The reviewers think this is an important and sound theoretical contribution. The authors should however make the changes suggested by the reviewers, for example, by adding remarks and discussions to make the theoretical results more understandable, and by clarifying differences from the existing work in [17].